# Suppressive Effects of β-Hydroxybutyrate Administration on Lipopolysaccharide-Induced Inflammation in Broiler Chickens

**DOI:** 10.3390/vetsci11090405

**Published:** 2024-09-02

**Authors:** Tae Horiuchi, Kyohei Furukawa, Motoi Kikusato

**Affiliations:** 1Laboratory of Animal Nutrition, Graduate School of Agricultural Science, Tohoku University, Aramaki Aza-Aoba, Sendai 980-8572, Japanfurukawa@agr.nagoya-u.ac.jp (K.F.); 2Laboratory of Animal Nutrition, Graduate School of Bioagricultural Sciences, Nagoya University, Furo-cho, Chikusa-ku, Nagoya 464-8601, Japan

**Keywords:** spleen, peripheral blood monocytes, ketolytic enzyme, inflammatory cytokine

## Abstract

**Simple Summary:**

β-hydroxybutyrate (BHB) is generated as one of the ketone bodies in the liver under fasted and hypoglycemic conditions. While BHB is utilized as an alternative energy substrate in extrahepatic tissues/organs, this ketone body has been reported to exert anti-inflammatory effects via modulating inflammatory signaling cascades. Several bacterial pathogens that originated in hygienic litter conditions consecutively threaten chickens. Lipopolysaccharides, a harmful component of Gram-negative bacteria, cause intestinal and systemic inflammation. Therefore, the management of inflammation is a significant concern in poultry production. However, the therapeutic effects of BHB on inflammation have yet to be investigated in chickens. This study demonstrated that BHB administration can alleviate LPS-induced inflammation in broiler chickens, and the effects could be dependent on the gene expression levels of ketolytic enzymes, BHB dehydrogenase-1 and succinyl-CoA:3-ketoacid CoA transferase, in extrahepatic tissues/organs. This study is the first to show the therapeutic effects of BHB on LPS-induced inflammation in chickens, possibly via the involvement of BHB utilization.

**Abstract:**

Background: This study aimed to evaluate the suppressive effects of β-hydroxybutyrate (BHB) administration on lipopolysaccharide (LPS)-induced inflammation in broiler chickens. Methods: Twenty-day-old male broiler chickens were randomly allocated to three groups, each of which was treated with saline (control), intraperitoneal administration of LPS [1.5 mg/kg body weight (BW), *Escherichia coli* O127:B8], or LPS plus BHB (3 mmol/kg BW). Results: Plasma albumin and total protein concentration were significantly reduced by LPS administration, while BHB co-treatment partially attenuated the effects. The LPS treatment significantly induced plasma aspartate and alanine aminotransferase activities, and interleukin (IL)-6 concentration, with the increases suppressed by BHB co-treatment (*p* < 0.05). The LPS treatment significantly increased the gene expression levels of IL-1β, IL-6, and IL-18 in the spleen and peripheral blood monocytes (PBMC), while the increases were partially attenuated by BHB in the spleen. Relatively higher levels of BHB dehydrogenase 1 and succinyl-CoA:3-ketoacid CoA transferase were observed in the spleen and skeletal muscle, while these gene levels were lower in PBMC and the liver. Conclusions: The present results suggest that BHB can suppress LPS-induced inflammation, in which ketolytic enzyme expression levels may be involved in broiler chickens.

## 1. Introduction

Ketone bodies, acetoacetate, acetone, and β-hydroxybutyrate (BHB) are water-soluble substrates generated in the liver under fasted and hypoglycemic conditions. In the ketogenic process, acetyl-CoA derived from fatty acids is converted into acetoacetyl-CoA, from which acetoacetate is generated with 3-hydroxy-3-methylglutaryl-CoA synthase (HMGCS) and 3-hydroxy-3-methylglutaryl-CoA lyase (HMGCL) (Figure 1). Thereafter, BHB is generated with BHB dehydrogenase (BDH), with acetone also generated by non-enzymatic decarboxylation. In extrahepatic tissues/organs, BHB is metabolized to acetoacetate, and this compound is subsequently re-converted into acetoacetyl-CoA, followed by acetyl-CoA formation. These ketolytic processes contribute to ATP production in mitochondria as an alternative energy source. In post-hatch chicks, plasma BHB concentration is higher as it is generated from residual egg yolk. The BHB levels are dramatically reduced with time after hatching [1,2]. It has also been reported that serum BHB concentration is not changed by 12 h of feed withdrawal, while the level is increased by 24 h treatment and maintained constantly up to the next 24 h in young chickens [3]. These reports suggest that BHB could be used as an energy substrate in response to physiological conditions.

Apart from the properties of BHB as energy fuel, recent studies have demonstrated that BHB exerts inflammatory effects due to a modulation of the signaling cascade [4,5]. It has been reported that BHB induces forkhead box protein O1 and its target gene, heme oxygenase-1 gene expression [6], reinforcing the anti-inflammatory effect of interleukin (IL)-10 [7]. Moreover, BHB has been reported to block the formation and activation of NLR family pyrin domain containing 3 (NLRP3) inflammasome [8,9,10] and stimulate GPR109A receptor [11], promoting anti-inflammation. Moreover, BHB administration has been reported to ameliorate renal inflammation, in which nuclear factor-erythroid 2 related factor 2 (Nrf2), a master regulator of antioxidative gene transcription [12], was activated with enhanced metabolic flux of TCA intermediates, acetoacetate, succinate, fumarate [13]. These lines of evidence suggest that BHB alleviates inflammation through various molecular signaling transduction pathways, and metabolic alterations could also be involved in the effects.

The administration of lipopolysaccharide (LPS), a cell wall constituent of Gram-negative bacteria, is often used as a pathogenic inflammation model [14,15,16,17]. However, there is no available information on the effects of BHB on the innate immune response of LPS-treated chickens. Therefore, the present study aimed to investigate the therapeutic effects of BHB on LPS-treated chickens by measuring the plasma inflammatory parameters and cytokine expression. The study also examined the gene expression of ketolytic enzymes to seek a possible mechanism exhibiting the BHB effects.

## 2. Materials and Methods

### 2.1. Animals and Experimental Design

Twenty-day-old Ross 308 male broiler chickens (*Gallus gallus domesticus*) were obtained from a local commercial hatchery (Matsumoto Poultry Farms & Hatcheries Co., Ltd., Zao, Miyagi, Japan). The chicks were bred according to the breeding manuals and were provided ad libitum access to water and feed, a corn/soybean-based standard diet for broiler chickens at the grower phase (crude protein, 22%; metabolizable energy, 3200 kcal/kg) until they were 25 days old. The chickens were randomly allocated to the following treatment groups with similar average body weight (BW): sterile 0.9% (*w*/*v*) sodium chloride solution (saline, control, n = 7), LPS [1.5 mg/kg (BW), *Escherichia coli* O127:B8 (#L3129; Sigma-Aldrich, St. Louis, MO, USA), n = 6], or LPS plus BHB sodium [3 mmol/kg BW (#H0231; Tokyo Chemical industry, Co., Ltd., Tokyo, Japan), n = 7]. The LPS and BHB solutions were prepared using saline on the day of use. The chickens were intraperitoneally injected with BHB for 3 h, with LPS injected for 2 h before euthanasia. The same aliquot of saline was injected for treatments without LPS or BHB. Feed was withdrawn in all groups during the treatment. Chickens were euthanized by decapitation, and the spleen, liver, and *gastrocnemius* muscles were then excised and immediately frozen/powdered in liquid nitrogen. For the isolation of PBMC, whole blood was collected in a heparinized centrifuge tube from the wing vein. The blood was gently transferred onto Lymphoprep™ solution (#ST-07811; STEMCELL Technologies, Vancouver, BC, Canada) and thereafter centrifuged at 800× *g* for 30 min at 20 °C. The organs, tissues, and PBMC were stored at −80 °C until use.

### 2.2. Analyses of Plasma Inflammation Markers

Plasma was obtained from heparinized-whole blood by centrifugation at 825× *g* for 10 min at 4 °C. The following inflammation markers were measured using each commercial kit: aspartate aminotransferase (AST) and alanine aminotransferase (ALT) activities (#431-30901; FUJIFILM-Wako Pure Chemical Corporation, Osaka, Japan), and albumin and total protein concentration (#274-24301; FUJIFILM-Wako Pure Chemical Corporation, Osaka, Japan). The study also measured plasma interleukin (IL)-6 concentration using a commercial kit (#MBS2021018; MyBioSource, Inc., San Diego, CA, USA) according to the manufacturer’s instructions.

### 2.3. Quantification of Gene Expression Levels

Real-time reverse transcript polymerase chain reaction (RT-PCR) was performed to quantify the gene expression levels of the inflammatory cytokines and ketogenic and ketolytic enzymes. Tissue RNA was isolated from the spleen, PBMC, skeletal muscle, and liver. Synthesis of complementary DNA and real-time RT-PCR analysis were conducted as previously described [18,19]. Inflammatory cytokines (IL-1β, IL-6, IL-18), and ketogenic and ketolytic enzymes (HMGCL, HMGCS2, BDH1, SCOT), were amplified using a specific primer (Table 1). Amplification was performed using CFX Connect^®^ Real-Time System (Bio-Rad Laboratories, Inc., Hercules, CA, USA) with the following cycling conditions: an initial denaturation step at 95 °C for 3 min, followed by 40 cycles consisting of 10 s at 95 °C for DNA denaturation, 10 s for primer annealing (see the temperatures in Table 1), and 30 s at 72 °C for DNA extension. The values were normalized to the expression levels of ribosomal protein S9 (RPS9) as an internal standard.

### 2.4. Statistical Analysis

Data are presented as the mean ± standard error (SE) of 6–7 individuals. Statistical analysis was performed using one-way analysis of variance with Bonferroni’s method, with values of *p* < 0.05 indicating statistical significance in each test.

## 3. Results

### 3.1. Plasma Inflammatory Parameters

Plasma albumin and total protein concentration were significantly reduced by LPS administration, while the effects were partially inhibited in the BHB co-treated group (*p* < 0.05 in albumin; *p* = 0.07 in total protein) (Figure 2A,B). The LPS treatment significantly increased plasma AST and ALT activities, and the increases did not occur in the BHB co-treated group (Figure 2C,D), and similar results were obtained in plasma IL-6 concentration (Figure 2E). These results suggest that BHB intraperitoneal administration may have a suppressive effect on LPS-induced acute inflammation in broiler chickens.

### 3.2. The Effects of LPS and BHB Administration on Inflammatory Gene Expression

The gene expression levels of inflammatory cytokines were measured. As illustrated in Figure 3, the IL-1β, IL-6, and IL-18 gene expression levels were significantly increased by LPS administration in the spleen and PBMC, with greater up-regulation observed in the spleen than in PBMC. The above changes were partially inhibited in the BHB co-treated group in the spleen (*p* < 0.05); however, the suppression did not occur in PBMC (Figure 3).

### 3.3. Different Gene Expression of Ketogenic and Ketolytic Enzymes in Peripheral Tissues/Organs

Beta-hydroxybutyrate is metabolized to acetyl-CoA (Figure 1) and subsequently yields TCA-cycle intermediates. Therefore, the present study examined the gene expression levels of the enzymes of ketone body metabolism to seek possible machinery associated with the different responses of BHB effects on LPS-induced inflammation between the spleen and PBMC. The gene expression levels of HMGCS2 and HMGCL were investigated as a rate-limiting enzyme of ketone body synthesis, each of which catalyzes the formation of acetoacetyl-CoA and 3-hydroxy-3-methylglutaryl-CoA from acetyl-CoA and acetoacetyl-CoA, respectively. The study also measured the gene expression levels of BDH1 and SCOT, each of which catalyzes a reversible reaction of BHB to acetoacetate and acetoacetate to acetoacetyl-CoA, respectively. The above four gene expression levels were also measured in the liver and skeletal muscle tissue as typical ketogenetic and ketolytic tissues/organs, respectively.

The present study showed that both HMGCS2 and HMGCL gene levels were markedly higher in the liver compared to those of skeletal muscle, PBMC, and the spleen (^ab^
*p* < 0.01) (Figure 4A,B) since the liver is a major ketogenic organ. Next, as seen in Figure 4C,D, BDH1 and SCOT gene levels were relatively higher in skeletal muscle and the spleen because they use ketone bodies as an alternative energy substrate. Meanwhile, these gene expressions in PBMC were lower than the above extrahepatic tissues/organs and comparable to those observed in the liver. These results suggest that PBMC may be unable to utilize BHB, which could explain the little effect of BHB on the LPS-induced inflammatory response.

## 4. Discussion

There is little information regarding the effects of BHB on the innate immune response of chickens, to our knowledge. One study showed a possible involvement of BHB in chicken inflammation: serum BHB concentration was increased with ingestion of anti-inflammatory plant polysaccharides in pathogen-challenged laying hens [20]. This report suggests that BHB could participate in suppressing pathogen-induced inflammation, although it is not evident that BHB directly suppresses inflammation. Therefore, the present study is the first to demonstrate that BHB administration may alleviate LPS-induced inflammation in chickens. While the present study did not address the precise mechanism of the BHB effects, one could suggest that the expression levels of the ketolytic enzymes, BDH1 and SCOT, could participate in the occurrence of the therapeutic effects of BHB in chickens.

It has been reported that intraperitoneal BHB injection mitigates kidney and placental injury [9,21,22]. In the present study, BHB was administered intraperitoneally to avoid the interference of altered intestinal functions and microbial compositions to inflammatory status. Oral BHB administration was reported to alleviate the intestinal integrity of mice [23], and poly-BHB administration was reported to activate intestinal butyrate production, probably through the production of BHB and its proliferating effect on butyrate-producing bacteria [24]. From these findings, it could be considered that the oral BHB administration model could not precisely evaluate the BHB effects because other factors, such as short-chain fatty acid or the prevention of pathogen incorporation, may be associated with the inflammatory status. Thus, using an intraperitoneal BHB injection model, the present study indicates that BHB may directly contribute to improving inflammatory status.

The study found that the effects could depend on the gene expression levels of the ketolytic enzymes, BDH1 and SCOT. It has been reported that BDH1 plays a pivotal role in the suppression of diabetic kidney injury [13]. The study proposed the suppression machinery based on the interaction of metabolites with a transcriptional factor; fumarate yielded from BHB catabolism induces a nuclear translocation of Nrf2, suppressing inflammation/oxidative stress in the kidney. Given the findings, it could be considered that BDH1-meditated BHB metabolism participates in the BHB effects in chickens. It should be noted that the liver does not express BDH1 and SCOT. Therefore, it could be considered that BHB has little effect on hepatic inflammation. However, the present study showed that BHB alleviated inflammatory cytokine expression and ALT activity, which mainly depends on hepatic inflammatory/injury status, in LPS-treated chickens. A few studies have demonstrated that BHB administration alleviated hepatic inflammation in postnatal piglets exhibiting growth retardation [25] and human hepatocarcinoma HepG2 cells [26]. The latter report suggests that AMP-activated protein kinase activation may be involved in the anti-inflammatory effects. These findings suggest that the BHB effects on hepatic inflammation could be independent of the ketolytic enzymes.

The present and previous studies have shown the therapeutic effects of BHB in several inflammation models. However, a few investigations have reported the toxin-like or inflammatory effects of BHB. It has been reported that BHB exacerbates the LPS/d-galactosamine-induced inflammatory response in mice [27], and promotes inflammatory gene expression in calf hepatocytes [28]. Moreover, acetoacetate derived from BHB as one of the ketone bodies has been reported to trigger NLRP3 inflammasome activation in bovine PBMC. The administration dosage, exposure time, inflammation model, and animals used differed between the studies. Therefore, it could be difficult to obtain a consistent result for BHB effects, considering these reports and the present results. Increased ketone bodies in circulation due to metabolic disorders often induce ketoacidosis. From these lines of findings, it could be suggested that the physiological conditions prior to BHB administration are an important factor in exerting the therapeutic/anti-inflammatory effects of BHB.

Recent investigations have proposed that metabolic intermediates and enzymes potentiate anti-inflammation [29], and that ketogenesis favors oxidative phosphorylation to promote disease tolerance [30]. Therefore, BHB utilization may be one of the determinants of the anti-inflammatory effects. The present study found that isolated BHB had little effect on the mitochondrial respiration rate of isolated/cultured PBMC when the ketone body was supplemented into the cultivation medium as a major respiratory substrate, using the Seahorse Bioscience extracellular flux analyzer. This result could support the idea that chicken PBMC cannot metabolize BHB. The present study was primarily conducted at a minimal scale; however, the repeated trials exhibited almost similar results on the anti-inflammatory effects of BHB. Further investigation using other tissues/organs, such as the kidney, brain, and intestine, is needed to precisely evaluate the involvement of ketolytic enzymes in BHB effects. Nonetheless, the present study is the first to demonstrate that BHB alleviated LPS-induced inflammation, in which the gene expression of ketolytic enzymes could be involved in extrahepatic cells/organs in chickens.

## Figures and Tables

**Figure 1 vetsci-11-00405-f001:**
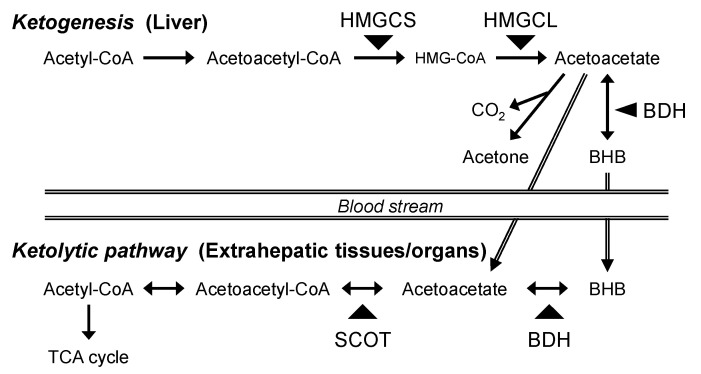
Schematic representation of ketogenic and ketolytic pathways.

**Figure 2 vetsci-11-00405-f002:**
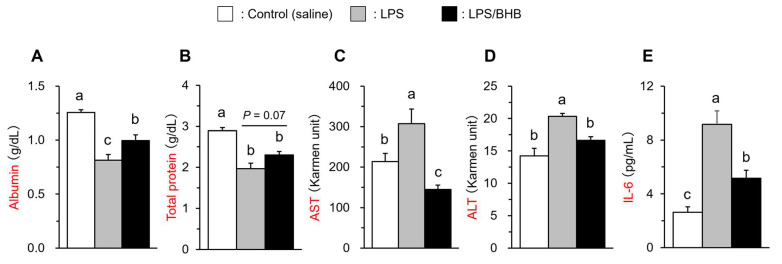
Effects of LPS and BHB on plasma albumin (**A**), total protein (**B**) concentrations, AST (**C**) and ALT (**D**) activities, IL-6 concentration (**E**) in broiler chickens. Data are expressed as means + SE, n = 6–7. ^abc^
*p* < 0.05 analyzed by Tukey–Kramer multiple comparison test, with different superscript letters indicating statistical difference.

**Figure 3 vetsci-11-00405-f003:**
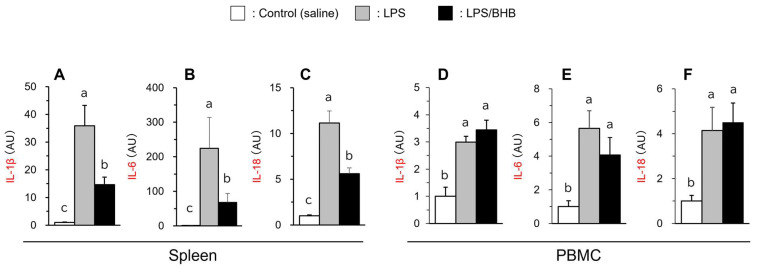
Effects of LPS and BHB on IL-1β (**A**,**D**), IL-6 (**B**,**E**), and IL-18 (**C**,**F**) in the spleen and PBMC of broiler chickens. Data are expressed as means + SE, n = 6–7. ^abc^ *p* < 0.05 analyzed by Tukey–Kramer multiple comparison test, with different superscript letters indicating statistical difference. Data are represented as fold changes relative to control values.

**Figure 4 vetsci-11-00405-f004:**
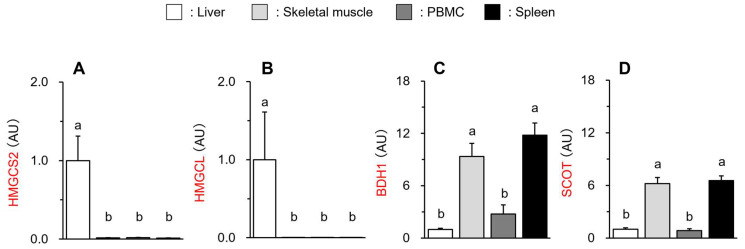
Different gene expression levels of HMGCS2 (**A**), HMGCL (**B**), BDH1 (**C**), and SCOT (**D**) of the liver, skeletal muscle, PBMC, and spleen of broiler chickens. Data are expressed as means + SE, n = 6–7. ^ab^
*p* < 0.01 analyzed by Tukey–Kramer multiple comparison test, with different superscript letters indicating statistical difference. Data are represented as fold changes relative to the values of the liver.

**Table 1 vetsci-11-00405-t001:** Primer sequences.

Gene Symbol	Primer Sequence (5′-3′) Sense (Upper); Antisense (Lower)		Annealing Temperature (°C)	Accession ID
IL-1β	Sense	CTGCCTGCAGAAGAAGCCT	59	NP_989855.1
	Antisense	ATGTCGAAGGACTGTGAGCG		
IL-6	Sense	CGACGAGGAGAAATGCCTGA	51	NP_989959.1
	Antisense	GGGATGACCACTTCATCGGG		
IL-18	Sense	TGATGAGCTGGAATGCGATGCC	59	NP_989939.1
	Antisense	TGGACGAACCACAAGCAACTGG		
HMGCL	Sense	GGTACTCCCACACTCTGGCG	58	NP_001185643.1
	Antisense	GGACGCATGAAACATACCCAC		
HMGCS2	Sense	AGTGGCAAAAAGAGGGGACA	56	XM_422225.8
	Antisense	CAGTCTTGCCACCGACTTCT		
BDH1	Sense	GGAGGTCAAAGGGTCGTGTA	56	NP_001006547.3
	Antisense	CAGGTTGGTGGCAGCTATGA		
SCOT	Sense	TCTACCAGCTGTCATCGCAA	63	NP_001006578.2
	Antisense	TCCAAAATTGTCAACGCCTGC		
RPS9	Sense	TGCGAAGTTTTGTGACTGAAACA	60	NM_001277757
	Antisense	ATTCTTGGAGCATTCAGCCTTTC		

Abbreviations: IL, interleukin; HMGCL, 3-hydroxy-3-methylglutaryl-CoA lyase; HMGCS2, 3-hydroxy-3-methylglutaryl-CoA synthase 2; BDH1, β-hydroxybutyrate dehydrogenase 1; SCOT, succinyl-CoA:3-ketoacid CoA transferase; RPS9, ribosomal protein S9.

## Data Availability

No new data were created or analyzed in this study. Data sharing is not applicable to this article.

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
