# Peer review of "Suppressive Effects of β-Hydroxybutyrate Administration on Lipopolysaccharide-Induced Inflammation in Broiler Chickens"

_vetsci, 2024, doi:10.3390/vetsci11090405_

Round 1

Reviewer 1 Report

Comments and Suggestions for Authors

Mr. Tae Horiuchi, I am pleased to review your manuscript. You have validated the inhibitory effect of β - Hydroxybutyrate on LPS induced inflammation through inflammation indicators and gene expression levels of ketogenic and ketolytic enzymes. However, minor amendments might be necessary prior to their publication in Veterinary Science. Here are the areas I think need to be modified:

Line13-15Lipopolysaccharides are found in Gram-negative bacteria, which logically conflicts with several bacterial pathogens you mentioned earlier. I suggest instead that lipopolysaccharides, a harmful component of gram-negative bacteria, cause intestinal and systemic inflammation.

Line94;220: Use “chickens” instead of birds.

Line127I suggest that it would be more appropriate to change the + to ±.xx

Line175You may have written it wrong, it should be Figures 4A and 4B.

Line216There are 2 Spaces between “effects” and “in”.

In animal experiments, the number of chickens in each group is only 6-7, which I think is too small and the results are not convincing. For example, in your results in Figure 3 and Figure 4, some results have relatively large error bars, which may be caused by too few samples in your experiment.Another result that struck me as odd was why the LPS/BHB group had a lower AST Kaman unit than the blank group in Figure 3C. Was it a data error or something else?

Comments on the Quality of English Language

Overall OK, a small part needs to be modified

Author Response

  1. Line 13-15: Lipopolysaccharides are found in Gram-negative bacteria, which logically conflicts with several bacterial pathogens you mentioned earlier. I suggest instead that lipopolysaccharides, a harmful component of gram-negative bacteria, cause intestinal and systemic inflammation.
  • Thank you for the expert comments. The sentence was revised as suggested.
  1. Line 94; 220: Use "chickens" instead of "birds."
  • The words were revised as suggested.
  1. Line 127: I suggest that it would be more appropriate to change the "+" to "±" .xx
  • The word was revised as suggested.
  1. Line 175: You may have written it wrong, it should be Figures 4A and 4B.
  • Thank you for the indication. The word was revised.
  1. Line 21: There are 2 Spaces between "effects" and "in."
  • Thank you for your indication. Unfortunately, we could not find the errors that may be shown in certain display devices and conditions. We will check them during the proof-reading process if accepted.
  1. In animal experiments, the number of chickens in each group is only 6-7, which I think is too small and the results are not convincing. For example, in your results in Figure 3 and Figure 4, some results have relatively large error bars, which may be caused by too few samples in your experiment. Another result that struck me as odd was why the LPS/BHB group had a lower AST Kaman unit than the blank group in Figure 3C. Was it a data error or something else?
  • Thank you for the expert comments. We performed the trial twice, and the displayed results were only from the first trial. Almost similar results were obtained from the second trial. As suggested, it is unclear why the plasma AST activity of LPS/BHB-treated chickens was lower than that of the non-treated chickens in the first trial. However, the activity was comparable to that of the LPS-treated group when the results of the first and second trials were combined and analyzed statistically. We considered presenting the combined data from all the trials; however, a few parameters were not assessed yet. Therefore, we showed the results with measurements completely done. for readability. A part of the above responses was added to the Discussion section (lines 257-259). Thank you for your consideration.

Reviewer 2 Report

Comments and Suggestions for Authors

Suppressive Effects of β-Hydroxybutyrate Administration on Lipopolysaccharide-Induced Inflammation in Broiler Chickens

Horiuchi, et. al., Veterinary Sciences

The authors present a study of the effects of BHB on reducing LPS induced inflammation in broiler chickens.  This study is of general interest to the field as little is known about the therapeutic effect of BHB in the model of interest, the broiler chicken.  It is of particular interest that the authors exposed to e. coli derived LPS, as chickens can often present commercially with e. coli infections, and thus the potential of BHB to treat these types of infections is an important area of exploration.  The greatest weakness of this study is that it fails to provide a high amount of insight into the molecular mechanisms linking BHB to reduction in inflammation.  It is also lacking in exploration of markers beyond mRNA, which can unfortunately not relay a great deal of information concerning protein levels or enzymatic activity.  It is recommended that enzymatic assays and/or western blots be conducted to strengthen the assertions of the authors.  However, the study in its present form is valuable in its ability to at least pave the way for more exploration of the value of BHB as an anti-inflammatory in chickens.  Several additional specific suggestions are listed below.

Simple Summary/Abstract: The opening statement concerning what BHB is and how it is produced should be stated in the abstract in addition to the simple summary in the background section.  This clarifies why it is important to study BHB in the abstract, as many scientific professionals may read the abstract without reading the simple summary.

Figures 1-4: Some description needs to be placed in the figure legend indicating what is meant by the lowercase a, b, and c placed over the bars in each graph.  This appears to indicated some statistically significant difference, but the difference is not explained.

Figure 4: The authors should report the ct values of the internal control, as these levels may have varied between tissues.  At a minimum, the authors should show through statistical analysis that these levels were not different between organs in order to compare expression levels of other genes.

Comments on the Quality of English Language

English language is proficient and only requires standard spell and grammar check.

Author Response

  1. The greatest weakness of this study is that it fails to provide a high amount of insight into the molecular mechanisms linking BHB to reduction in inflammation. It is also lacking in exploration of markers beyond mRNA, which can unfortunately not relay a great deal of information concerning protein levels or enzymatic activity. It is recommended that enzymatic assays and/or western blots be conducted to strengthen the assertions of the authors. However, the study in its present form is valuable in its ability to at least pave the way for more exploration of the value of BHB as an anti-inflammatory in chickens.
  • Thank you for the kind comments. As mentioned, the study was a pilot study, and the underlying mechanism needs to be clarified thereafter.  
  1. Simple Summary/Abstract: The opening statement concerning what BHB is and how it is produced should be stated in the abstract in addition to the simple summary in the background section. This clarifies why it is important to study BHB in the abstract, as many scientific professionals may read the abstract without reading the simple summary.
  • Thank you for the careful indication. The advice is very valuable, and we considered it. However, it was impossible because of the limitation of word numbers. Thank you for your understanding.
  1. Figures 1-4: Some description needs to be placed in the figure legend indicating what is meant by the lowercase a, b, and c placed over the bars in each graph. This appears to indicated some statistically significant difference, but the difference is not explained.
  • Thank you for the expert comments. The brief explanation was added to each figure legend for the meaning of the superscript letters.
  1. Figure 4: The authors should report the ct values of the internal control, as these levels may have varied between tissues. At a minimum, the authors should show through statistical analysis that these levels were not different between organs in order to compare expression levels of other genes.
  • Thank you for the expert comments. We confirmed almost similar and non-statistically significant expression levels of RPS9 gene, as an internal control, between the tissues/organs measured. For the analyses using PCR, the amplified products (ng/ml) were used as the standard to determine each target and inner control gene quantification.

Reviewer 3 Report

Comments and Suggestions for Authors

Comments for the Author (Required):

The study by Horiuchi T et al, titled “Suppressive Effects of β-Hydroxybutyrate Administration on Lipopolysaccharide-Induced Inflammation in Broiler Chickens aims to study the protective effect of β-Hydroxybutyrate on LPS-mediated inflammation. They have shown that LPS decreases plasma albumin and upregulates cytokine gene expression. However, β-Hydroxybutyrate suppresses LPS-mediated effects. I have some concerns about this article.

Page 3, Experimental design- The chickens were intraperitoneally injected with saline and LPS for 3 h, with BHB injected for 2 h before euthanasia- Was LPS and BHB co-injected together, or was LPS injected 1 h before BHB injection? Please clarify.

PCR cycling parameters should be furnished in the methods section.

Fig 2,3,4- Author should mention the P-valves/* in the respective figures.

Fig 2,3,4- It would be nice to mention the appropriate title for the y-axis in the respective figures.

Author should measure the cytokine levels in the plasma

Figure 4. Where are the control and LPS/BHB groups??

Figure 4. Author should measure the enzyme activity

Author Response

  1. Page 3, Experimental design- The chickens were intraperitoneally injected with saline and LPS for 3 h, with BHB injected for 2 h before euthanasia- Was LPS and BHB co-injected together, or was LPS injected 1 h before BHB injection? Please clarify.
  • Thank you for the expert comments. We apologize for the mistakes in the description regarding the injection time. Collectively, the chickens were intraperitoneally injected with BHB for 3 h, with saline and LPS injected for 2 h before euthanasia (line 94-95). We understood that the Reviewer wanted to know why the LPS and BHB did not inject simultaneously. To this point, there are a few reasons. First, it was to avoid the negative interaction between the treatments once they were injected simultaneously. Second, the time to circulate BHB injected was needed to exert the effects on several tissues/organs. The present study did not assess if the BHB effects were also observed when the LPS and BHB administration orders were reversed. The point is one of the future issues to be addressed. We will consider describing the above reasons in the manuscript if necessary. Thank you for your consideration.
  1. PCR cycling parameters should be furnished in the methods section.
  • As indicated, the cycling parameters and annealing temperature were added (lines 121-125).
  1. Fig 2,3,4: Author should mention the P-values/* in the respective figures.
  • We considered the request, but it is difficult to insert the p-values for each figure in terms of clear visuality. Statistical differences are shown as superscript letters (abcP < 0.05) which still be used generally. So, please see them carefully to identify the differences.
  1. Fig 2,3,4: It would be nice to mention the appropriate title for the y-axis in the respective figures.
  • As suggested, the parameter names were added as the y-axis for each figure.
  1. Author should measure the cytokine levels in the plasma.
  • We measured plasma IL-6 concentrations, whose result was illustrated in Figure 2E.
  1. Figure 4. Where are the control and LPS/BHB groups??
  • The data displayed in Figure 4 were only from the control group (non-treatment chickens). We did not assess the effects of LPS or BHB on gene expression levels of ketogenic and ketolytic genes; however, it is conceivable that their constant expression level could be involved in the difference in the BHB effect between the organs/tissues.
  1. Figure 4. Author should measure the enzyme activity.
  • Thank you for the expert comments. We have not yet found a reliable method to measure each enzyme's activity. However, the role of BHB as respiratory substrates in PBMC was verified using the Seahorse Bioscience extracellular flux analyzer (XFe24, Agilent Technologies, Inc.). As a result, the experiment showed that BHB had little effect on mitochondrial respiration rate when BHB was supplemented into the cultivation medium as a major respiratory substrate. The result could support the idea that chicken PBMC cannot metabolize BHB. The sentence was added to lines 253-257.

Round 2

Reviewer 3 Report

Comments and Suggestions for Authors

The manuscript is now suitable for publication.